# Break the Ceiling: Stronger Multi-scale Deep Graph Convolutional Networks

**Sitao Luan**[1,2,*], **Mingde Zhao**[1,2,*], **Xiao-Wen Chang**[1], **Doina Precup**[1,2,3]
{sitao.luan@mail, mingde.zhao@mail, chang@cs, dprecup@cs}.mcgill.ca
[1]McGill University; [2]Mila; [3]DeepMind

## Abstract

Recently, neural network based approaches have achieved significant progress for solving large, complex, graph-structured problems. Nevertheless, the advantages of multi-scale information and deep architectures have not been sufficiently exploited. In this paper, we first analyze key factors constraining the expressive power of existing Graph Convolutional Networks (GCNs), including the activation function and shallow learning mechanisms. Then, we generalize spectral graph convolution and deep GCN in block Krylov subspace forms, upon which we devise two architectures, both scalable in depth however making use of multi-scale information differently. On several node classification tasks, the proposed architectures achieve state-of-the-art performance.

## 1 Introduction & Motivation

Many real-world problems can be modeled as graphs [14, 18, 25, 12, 27, 7]. Inspired by the success of Convolutional Neural Networks (CNNs) [20] in computer vision [22], graph convolution defined on graph Fourier domain stands out as the key operator and one of the most powerful tools for using machine learning to solve graph problems. In this paper, we focus on spectrum-free Graph Convolutional Networks (GCNs) [2, 29], which have demonstrated state-of-the-art performance on many transductive and inductive learning tasks [7, 18, 25, 3, 4].

One major problem of the existing GCNs is the low expressive power limited by their shallow learning mechanisms [38, 36]. There are mainly two reasons why people have not yet achieved an architecture that is scalable in depth. First, this problem is difficult: considering graph convolution as a special form of Laplacian smoothing [21], networks with multiple convolutional layers will suffer from an over-smoothing problem that makes the representation of even distant nodes indistinguishable [38]. Second, some people think it is unnecessary: for example, [2] states that it is not necessary for the label information to totally traverse the entire graph and one can operate on the multi-scale coarsened input graph and obtain the same flow of information as GCNs with more layers. Acknowledging the difficulty, we hold on to the objective of deepening GCNs since the desired compositionality[1] will yield easy articulation and consistent performance for problems with different scales.

In this paper, we break the performance ceiling of the GCNs. First, we analyze the limits of the existing GCNs brought by the shallow learning mechanisms and the activation functions. Then, we show that any graph convolution with a well-defined analytic spectral filter can

be written as a product of a block Krylov matrix and a learnable parameter matrix in a special form. Based on this, we propose two GCN architectures that leverage multi-scale information in different ways and are scalable in depth, with stronger expressive powers and abilities to extract richer representations of graph-structured data. We also show that the equivalence of the two architectures can be achieved under certain conditions. For empirical validation, we test different instances of the proposed architectures on multiple node classification tasks. The results show that even the simplest instance of the architectures achieves state-of-the-art performance, and the complex ones achieve surprisingly higher performance, with or without validation sets.

## 2    Why Deep GCN Does Not Work Well?

### 2.1    Foundations

As in [11], we use bold font for vectors (*e.g.* $v$), block vectors (*e.g.* $V$) and matrix blocks (*e.g.* $V_i$). Suppose we have an undirected graph $\mathcal{G} = (\mathcal{V}, \mathcal{E}, A)$, where $\mathcal{V}$ is the node set with $|\mathcal{V}| = N$, $\mathcal{E}$ is the edge set with $|\mathcal{E}| = E$, $A \in \mathbb{R}^{N \times N}$ is a symmetric adjacency matrix and $D$ is a diagonal degree matrix, *i.e.* $D_{ii} = \sum_j A_{ij}$. A diffusion process [6, 5] on $\mathcal{G}$ can be defined by a diffusion operator $L$, which is a symmetric matrix, *e.g.* graph Laplacian $L = D - A$, normalized graph Laplacian $L = I - D^{-1/2}AD^{-1/2}$ and affinity matrix $L = A + I$, *etc.*. In this paper, we use $L$ for a general diffusion operator, unless specified otherwise. The eigendecomposition of $L$ gives us $L = U\Lambda U^T$, where $\Lambda$ is a diagonal matrix whose diagonal elements are eigenvalues and the columns of $U$ are the orthonormal eigenvectors, named graph Fourier basis. We also have a feature matrix (graph signals) $X \in \mathbb{R}^{N \times F}$ (which can be regarded as a block vector) defined on $\mathcal{V}$ and each node $i$ has a feature vector $X_{i,:}$, which is the $i$-th row of $X$.

Spectral graph convolution is defined in graph Fourier domain *s.t.* $x *_{\mathcal{G}} y = U((U^T x) \odot (U^T y))$, where $x, y \in \mathbb{R}^N$ and $\odot$ is the Hadamard product [7]. Following this definition, a graph signal $x$ filtered by $g_\theta$ can be written as

$$y = g_\theta(L)x = g_\theta(U\Lambda U^T)x = Ug_\theta(\Lambda)U^T x \qquad (1)$$

where $g_\theta$ is any function which is analytic inside a closed contour which encircles $\lambda(L)$, *e.g.* Chebyshev polynomial [7]. GCN generalizes this definition to signals with $F$ input channels and $O$ output channels and its network structure can be described as

$$Y = \text{softmax}(L \, \text{ReLU}(LXW_0) \, W_1) \qquad (2)$$

where

$$L \equiv \tilde{D}^{-1/2}\tilde{A}\tilde{D}^{-1/2}, \quad \tilde{A} \equiv A + I, \quad \tilde{D} \equiv \text{diag}(\sum_j \tilde{A}_{1j}, \dots, \sum_j \tilde{A}_{Nj}) \qquad (3)$$

This is called spectrum-free method [2] since it requires no explicit computation of eigendecomposition and operations on the frequency domain [38].

### 2.2    Problems

Suppose we deepen GCN in the same way as [18, 21], we have

$$Y = \text{softmax}(L \, \text{ReLU}(\cdots L \, \text{ReLU}(L \, \text{ReLU}(LXW_0) \, W_1) \, W_2 \cdots) \, W_n) \equiv \text{softmax}(Y') \qquad (4)$$

For this architecture, [21] gives an analysis on the effect of $L$ without considering the ReLU activation function. Our analyses on (4) can be summarized in the following theorems.

**Theorem 1.** Suppose that $\mathcal{G}$ has $k$ connected components and the diffusion operator $L$ is defined as that in (3). Let $X \in \mathbb{R}^{N \times F}$ be any block vector and let $W_j$ be any non-negative parameter matrix with $\|W_j\|_2 \leq 1$ for $j = 0, 1, \dots$. If $\mathcal{G}$ has no bipartite components, then in (4), as $n \to \infty$, $\text{rank}(Y') \leq k$.

*Proof*    See Appendix A.    □

**Conjecture 1.** Theorem 1 still holds without the non-negative constraint on the parameter matrices.

**Theorem 2.** Suppose the $n$-dimensional $x$ and $y$ are independently sampled from a continuous distribution and the activation function $\text{Tanh}(z) = \frac{e^z - e^{-z}}{e^z + e^{-z}}$ is applied to $[x, y]$ pointwisely, then

$$\mathbb{P}(\text{rank}\,(\text{Tanh}([x, y])) = \text{rank}([x, y])) = 1$$

*Proof*  See Appendix A.  □

Theorem 1 shows that if we simply deepen GCN, the extracted features will degrade, *i.e.* $Y'$ only contains the stationary information of the graph structure and loses all the local information in node for being smoothed. In addition, from the proof we see that the pointwise ReLU transformation is a conspirator. Theorem 2 tells us that Tanh is better at keeping linear independence among column features. We design a numerical experiment on synthetic data (see Appendix) to test, under a 100-layer GCN architecture, how activation functions affect the rank of the output in each hidden layer during the feedforward process. As Figure 1(a) shows, the rank of hidden features decreases rapidly with ReLU, while having little fluctuation under Tanh, and even the identity function performs better than ReLU (see Appendix for more comparisons). So we propose to replace ReLU by Tanh.

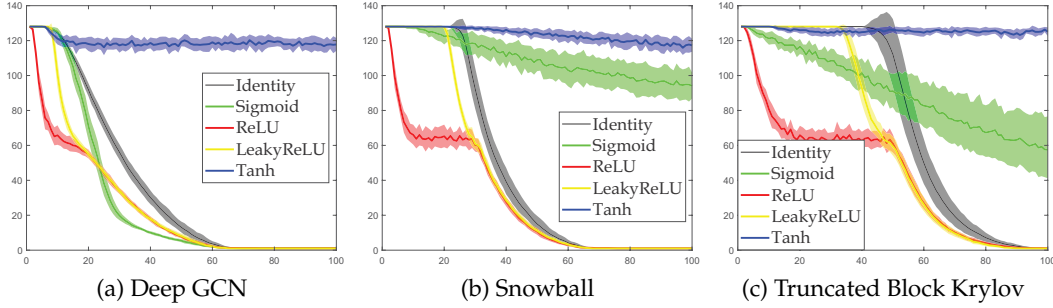

(a) Deep GCN        (b) Snowball        (c) Truncated Block Krylov

Figure 1: Changes in the number of independent features with the increment of network depth

## 3   Spectral Graph Convolution and Block Krylov Subspaces

### 3.1   Block Krylov Subspaces

Let $\mathbb{S}$ be a vector subspace of $\mathbb{R}^{F \times F}$ containing the identity matrix $I_F$ that is closed under matrix multiplication and transposition. We define an inner product $\langle \cdot, \cdot \rangle_\mathbb{S}$ in the block vector space $\mathbb{R}^{N \times F}$ as follows [11]:

**Definition 1** *A mapping $\langle \cdot, \cdot \rangle_\mathbb{S}$ from $\mathbb{R}^{N \times F} \times \mathbb{R}^{N \times F}$ to $\mathbb{S}$ is called a block inner product onto $\mathbb{S}$ if $\forall X, Y, Z \in \mathbb{R}^{N \times F}$ and $\forall C \in \mathbb{S}$:*

*1.  $\mathbb{S}$-linearity: $\langle X, YC \rangle_\mathbb{S} = \langle X, Y \rangle_\mathbb{S} C$ and $\langle X + Y, Z \rangle_\mathbb{S} = \langle X, Z \rangle_\mathbb{S} + \langle Y, Z \rangle_\mathbb{S}$*

*2.  symmetry: $\langle X, Y \rangle_\mathbb{S} = \langle Y, X \rangle_\mathbb{S}^T$*

*3.  definiteness: $\langle X, X \rangle_\mathbb{S}$ is positive definite if $X$ has full rank, and $\langle X, X \rangle_\mathbb{S} = 0_F$ iff $X = 0$.*

There are mainly three ways to define $\langle \cdot, \cdot \rangle_\mathbb{S}$ [11]: 1) (Classical.) $\mathbb{S}^{\text{Cl}} = \mathbb{R}^{F \times F}$ and $\langle X, Y \rangle_\mathbb{S}^{\text{Cl}} = X^T Y$; 2) (Global.) $\mathbb{S}^{\text{Gl}} = cI_F$, $c \in \mathbb{R}$ and $\langle X, Y \rangle_\mathbb{S}^{\text{Gl}} = \text{trace}(X^T Y)I_F$; 3) (Loop-interchange.) $\mathbb{S}^{\text{Li}}$ is the set of diagonal matrices and $\langle X, Y \rangle_\mathbb{S}^{\text{Li}} = \text{diag}(X^T Y)$. The three definitions are all useful yet we will use the classical one for our contribution.

For further explanations, we give the definition of block vector subspace of $\mathbb{R}^{N \times F}$.

**Definition 2** *Given a set of block vectors* $\{X_k\}_{k=1}^m \subset \mathbb{R}^{N \times F}$, *the* $\mathbb{S}$-*span of* $\{X_k\}_{k=1}^m$ *is defined as*
$$\text{span}^{\mathbb{S}}\{X_1, \ldots, X_m\} := \{\sum_{k=1}^m X_k C_k : C_k \in \mathbb{S}\}$$

Given the above definition, the order-$m$ block Krylov subspace with respect to the matrix $A \in \mathbb{R}^{N \times N}$, the block vector $B \in \mathbb{R}^{N \times F}$ and the vector space $\mathbb{S}$ can be defined as $\mathcal{K}_m^{\mathbb{S}}(A, B) :=$ $\text{span}^{\mathbb{S}}\{B, AB, \ldots, A^{m-1}B\}$. The corresponding block Krylov matrix is defined as $K_m(A, B) := [B, AB, \ldots, A^{m-1}B]$.

### 3.2 Spectral Graph Convolution in Block Krylov Subspace Form

In this section, we show that any graph convolution with well-defined analytic spectral filter defined on $L \in \mathbb{R}^{N \times N}$ can be written as the product of a block Krylov matrix with a learnable parameter matrix in a specific form. We take $\mathbb{S} = \mathbb{S}^{\text{Cl}} = \mathbb{R}^{F \times F}$.

For any real analytic scalar function $g$, its power series expansion around center 0 is
$$g(x) = \sum_{n=0}^{\infty} a_n x^n = \sum_{n=0}^{\infty} \frac{g^{(n)}(0)}{n!} x^n, \ |x| < R$$
where $R$ is the radius of convergence.

The function $g$ can be used to define a filter. Let $\rho(L)$ denote the spectrum radius of $L$ and suppose $\rho(L) < R$. The spectral filter $g(L) \in \mathbb{R}^{N \times N}$ can be defined as
$$g(L) := \sum_{n=0}^{\infty} a_n L^n = \sum_{n=0}^{\infty} \frac{g^{(n)}(0)}{n!} L^n, \ \rho(L) < R$$

According to the definition of spectral graph convolution in (1), graph signal $X$ is filtered by $g(L)$ as follows,
$$g(L)X = \sum_{n=0}^{\infty} \frac{g^{(n)}(0)}{n!} L^n X = \left[X, LX, L^2X, \cdots\right] \left[\frac{g^{(0)}(0)}{0!} I_F, \frac{g^{(1)}(0)}{1!} I_F, \frac{g^{(2)}(0)}{2!} I_F, \cdots\right]^T = A'B'$$
where $A' \in \mathbb{R}^{N \times \infty}$ and $B' \in \mathbb{R}^{\infty \times F}$. We can see that $A'$ is a block Krylov matrix and $\text{Range}(A'B') \subseteq \text{Range}(A')$. It is shown in [13, 11] that for $\mathbb{S} = \mathbb{R}^{F \times F}$ there exists a smallest $m$ such that
$$\text{span}^{\mathbb{S}}\{X, LX, L^2X, \cdots\} = \text{span}^{\mathbb{S}}\{X, LX, L^2X, \ldots, L^{m-1}X\} \tag{5}$$
where $m$ depends on $L$ and $X$ and will be written as $m(L, X)$ later. This means for any $k \geq m$, $L^k X \in \mathcal{K}_m^{\mathbb{S}}(L, X)$. From (5), the convolution can be written as
$$g(L)X = \sum_{n=0}^{\infty} \frac{g^{(n)}(0)}{n!} L^n X \equiv \left[X, LX, \ldots, L^{m-1}X\right]\left[(\Gamma_0^{\mathbb{S}})^T, (\Gamma_1^{\mathbb{S}})^T, \cdots, (\Gamma_{m-1}^{\mathbb{S}})^T\right]^T \equiv K_m(L, X)\Gamma^{\mathbb{S}} \tag{6}$$
where $\Gamma_i^{\mathbb{S}} \in \mathbb{R}^{F \times F}$ for $i = 1, \ldots, m-1$ are parameter matrix blocks. Then, a graph convolutional layer can be be generally written as
$$g(L)XW' = K_m(L, X)\Gamma^{\mathbb{S}}W' = K_m(L, X)W^{\mathbb{S}} \tag{7}$$
where $W^{\mathbb{S}} \equiv \Gamma^{\mathbb{S}}W' \in \mathbb{R}^{mF \times O}$. The essential number of learnable parameters is $mF \times O$.

### 3.3 Deep GCN in the Block Krylov Subspace Form

Since the spectral graph convolution can be simplified as (6)(7), we can build deep GCN in the following way.

Suppose that we have a sequence of analytic spectral filters $G = \{g_0, g_1, \ldots, g_n\}$ and a sequence of pointwise nonlinear activation functions $H = \{h_0, h_1, \ldots, h_n\}$. Then, a deep spectral graph convolution network can be written as
$$\mathbf{Y} = \text{softmax}\left\{g_n(L)\, h_{n-1}\left\{\cdots g_2(L)\, h_1\left\{g_1(L)\, h_0\left\{g_0(L)XW_0'\right\}W_1'\right\}W_2'\cdots\right\}W_n'\right\} \tag{8}$$

Define

$$H_0 = X, \qquad H_{i+1} = h_i\{g_i(L)H_iW_i\}, \;\; i = 0, \ldots, n-1$$

Then, we have

$$Y = \text{softmax}\{K_{m_n}(L, H_n)W_n^{\mathcal{S}_n}\}$$

From (7) and (8), we see we can write

$$H_{i+1} = h_i\{K_{m_i}(L, H_i)W_i^{\mathcal{S}_i}\}, \;\; m_i \equiv m(L, H_i)$$

It is easy to see that, when $g_i(L) = I$, (8) is a fully connected network [21]; when $n = 1$, $g_0(L) = g_1(L) = L$, where $L$ is defined in (3), it is just GCN [18]; when $g_i(L)$ is defined by the Chebyshev polynomial [15], $W_i' = I$, (8) is ChebNet [7].

### 3.4 Difficulties & Inspirations

In the last subsection, we gave a general form of deep GCN in the block Krylov form. Following this idea, we can leverage the existing block Lanczos algorithm [11, 10] to find $m_i$ and compute orthogonal basis of $\mathcal{K}_{m_i}^{\mathcal{S}}(L, H_i)$ which makes the filter coefficients compact [25] and improve numerical stability. But there are some difficulties in practice:

1. During the training phase, $H_i$ changes every time when parameters are updated. This makes $m_i$ a variable and thus requires adaptive size for parameter matrices $W_i^{\mathcal{S}_i}$.

2. For classical inner product, the $QR$ factorization that is needed in block Lanczos algorithm [11] is difficult to be implemented in backpropagation framework.

Despite implementation intractability, block Krylov form is still meaningful for constructing GCNs that are scalable in depth as we illustrate below.

For each node $v \in \{1, \ldots, N\}$ in the graph, denote $N(v)$ as the set of its neighbors and $N^k(v)$ as the set of its $k$-hop neighbors. Then, $LX(v,:)$ can be interpreted as a weighted mean of the feature vectors of $v$ and $N(v)$. If the network goes deep as (4), $Y'(v,:)$ becomes the "weighted mean" of the feature vectors of $v$ and $N^{(n+1)}(v)$ (not exactly weighted mean because we have ReLU in each layer). As the scope grows, the nodes in the same connected component tend to have the same (global) features, while losing their individual (local) features, which makes them indistinguishable. Such phenomenon is recognized as "oversmoothing" [21]. Though it is reasonable to assume that the nodes in the same cluster share many similar properties, it will be harmful to omit the individual differences between each node.

Therefore, the inspiration from the block Krylov form is that, to get a richer representation of each node, we need to concatenate the multi-scale information (local and global) together instead of merely doing smoothing in each hidden layer. If we have a smart way to stack multi-scale information, the network will be scalable in depth. To this end, we naturally come up with a densely connected architecture [17], which we call *snowball* network and a compact architecture, which we call the *truncated Krylov* network, in which the multi-scale information is used differently.

## 4 Deep GCN Architectures

### 4.1 Snowball

The block Krylov form inspires first an architecture that concatenates multi-scale features incrementally, resulting in a densely-connected graph network (Figure 2(a)) as follows:

$$
\begin{aligned}
&H_0 = X, \;\; H_{l+1} = f\left(L\left[H_0, H_1, \ldots, H_l\right]W_l\right), \;\; l = 0, 1, \ldots, n-1 \\
&C = g\left(\left[H_0, H_1, \ldots, H_n\right]W_n\right) \\
&\text{output} = \text{softmax}\left(L^p C W_C\right)
\end{aligned}
\qquad (9)
$$

where $W_l \in \mathbb{R}^{\left(\sum_{i=0}^l F_i\right) \times F_{l+1}}$, $W_n \in \mathbb{R}^{\left(\sum_{i=0}^n F_i\right) \times F_C}$ and $W_C \in \mathbb{R}^{F_C \times F_O}$ are learnable parameter matrices, $F_{l+1}$ is the number of output channels in layer $l$; $f$ and $g$ are pointwise activation functions;

$H_l$ are extracted features; $C$ is the output of a classifier of any kind, *e.g.*, a fully connected neural network or even an identity layer, in which case $C = [H_0, H_1, \ldots, H_n]$; $p \in \{0, 1\}$. When $p = 0$, $L^p = I$ and when $p = 1$, $L^P = L$, which means that we project $C$ back onto graph Fourier basis, which is necessary when the graph structure encodes much information. Following this construction, we can stack all learned features as the input of the subsequent hidden layer, which is an efficient way to concatenate multi-scale information. The size of input will grow like a snowball and this construction is similar to DenseNet [17], which is designed for regular grids (images). Thus, some advantages of DenseNet are naturally inherited, *e.g.*, alleviate the vanishing-gradient problem, encourage feature reuse, increase the variation of input for each hidden layer, reduce the number of parameters, strengthen feature propagation and improve model compactness.

## 4.2 Truncated Krylov

The block Krylov form inspires then an architecture that concatenates multi-scale features directly together in each layer. However, as stated in Section 3.4, the fact that $m_i$ is a variable makes GCN difficult to be merged into the block Krylov framework. Thus we compromise and set $m_i$ as a hyperparameter and get a truncated block Krylov network (Figure 2(b)) as shown below:

$$H_0 = X, \quad H_{l+1} = f\left(\left[H_l, LH_l \ldots, L^{m_l-1}H_l\right]W_l\right), \quad l = 0, 1, \ldots, n-1$$
$$C = g(H_n W_n) \tag{10}$$
$$\text{output} = \text{softmax}(L^p C W_C)$$

where $W_l \in \mathbb{R}^{(m_l F_l) \times F_{l+1}}$, $W_n \in \mathbb{R}^{F_n \times F_C}$ and $W_C \in \mathbb{R}^{F_C \times F_O}$ are learnable parameter matrices; $f$ and $g$ are activation functions; $C$ is the output of a classifier of any kind; $p \in \{0, 1\}$. In the truncated Krylov network, the local information will not be diluted in each layer because in each layer $l$, we start the concatenation from $L^0 H_l$ so that the extracted local information can be kept.

There are works on the analysis of error bounds of doing truncation in block Krylov methods [11]. But the results need many assumptions either on $X$, *e.g.*, $X$ is a standard Gaussian matrix [34], or on $L$, *e.g.*, some conditions on the smallest and largest eigenvalues of $L$ have to be satisfied [28]. Instead of doing truncation for a specific function or a fixed $X$, we are dealing with variable $X$ during training. So we cannot get a practical error bound since we cannot put any restriction on $X$ and its relation to $L$.

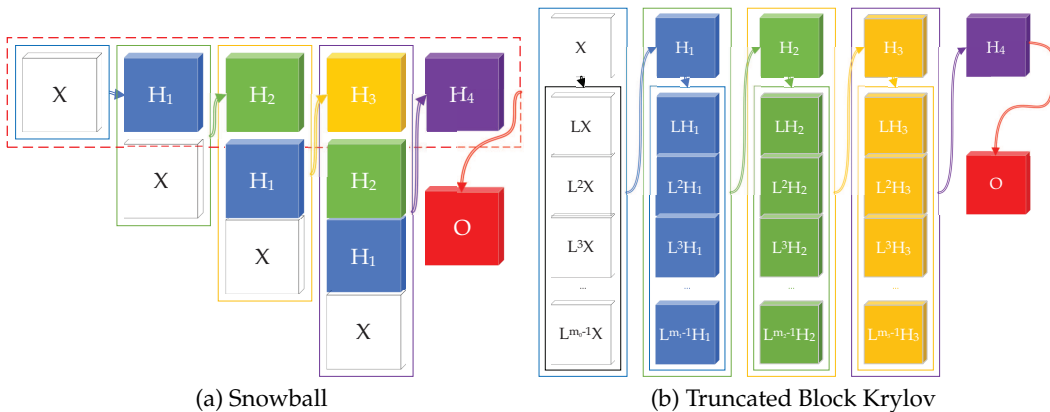

(a) Snowball            (b) Truncated Block Krylov

Figure 2: Proposed Architectures

The Krylov subspace methods are often associated with low-rank approximation methods for large sparse matrices. Here we would like to mention [25] does low-rank approximation of $L$ by the Lanczos algorithm. It suffers from the tradeoff between accuracy and efficiency: the information in $L$ will be lost if $L$ is not low-rank, while keeping more information via increasing the Lanczos steps will hurt the efficiency. Since most of the graphs we are dealing

with have sparse connectivity structures, they are actually not low-rank, *e.g.*, the Erdős-Rényi graph $G(n, p)$ with $p = \omega(\frac{1}{n})$ [32] and examples in Appendix IV. Thus, we do not propose to do low-rank approximation in our architecture.

## 4.3 Equivalence of Linear Snowball GCN and Truncated Block Krylov Network

In this part, we will show that the two proposed architectures are inherently connected. In fact their equivalence can be established when using identify functions as $f$, identity layer as $C$ and constraining the parameter matrix of truncated Krylov to be in a special form.

In linear snowball GCN, we can split the parameter matrix $W_i$ into $i + 1$ blocks and write it as $W_i = \left[(W_i^{(1)})^T, \cdots, (W_i^{(i+1)})^T\right]^T$ and then following (9) we have

$$H_0 = X, \; H_1 = LXW_0, \; H_2 = L[X, H_1]W_1 = LXW_1^{(1)} + L^2 XW_0^{(1)} W_1^{(2)} = L[X, LX]\begin{bmatrix} I & 0 \\ 0 & W_0^{(1)} \end{bmatrix}\begin{bmatrix} W_1^{(1)} \\ W_1^{(2)} \end{bmatrix}, \; \cdots$$

As in (9), we have $CW_C = L[H_0, H_1, \dots, H_n]W_C$. Thus we can write

$$[H_0, H_1 \cdots, H_n]W_C$$
$$= [X, LX, \cdots, L^n X]\begin{bmatrix} I & 0 & \cdots & 0 \\ 0 & I & \cdots & 0 \\ \vdots & \vdots & \ddots & \vdots \\ 0 & 0 & \cdots & W_0^{(1)} \end{bmatrix}\begin{bmatrix} I & 0 & \cdots & 0 \\ 0 & I & \cdots & 0 \\ \vdots & \vdots & \ddots & \vdots \\ 0 & 0 & \cdots & W_1^{(1)} \end{bmatrix}\cdots\begin{bmatrix} I & 0 & \cdots & 0 \\ 0 & W_{n-1}^{(n)} & \cdots & 0 \\ \vdots & \vdots & \ddots & \vdots \\ 0 & 0 & \cdots & W_{n-1}^{(1)} \end{bmatrix}\begin{bmatrix} W_C^{(1)} \\ W_C^{(2)} \\ \vdots \\ W_C^{(n)} \end{bmatrix}$$

which is in the form of (7), where the parameter matrix is the multiplication of a sequence of block diagonal matrices whose entries consist of identity blocks and blocks from other parameter matrices. Though the two proposed architectures stack multi-scale information in different ways, *i.e.* incremental and direct respectively, the equivalence reveals that the truncated block Krylov network can be constrained to leverage multi-scale information in a way similar to the snowball architecture. While it is worth noting that when there are no constraints, truncated Krylov is capable of achieving more than what snowball does.

## 4.4 Relation to Message Passing Framework

We denote the concatenation operator as $\|$. If we consider $L$ as a general aggregation operator which aggregates node features with its neighborhood features, we see that the two proposed architectures both have close relationships with message passing framework [12], which are illustrated in the following table, where $N^0(v) = \{v\}$, $M_t$ is a message function, $U_t$ is a vertex update function, $m_v^{(t+1)}, h_v^{(t+1)}$ are messages and hidden states at each node respectively, $m^{(t+1)} = [m_1^{(t+1)}, \cdots, m_N^{(t+1)}]^T$, $h^{(t+1)} = [h_1^{(t+1)}, \cdots, h_N^{(t+1)}]^T$ and $\sigma$ is a nonlinear activation function.

Compared to our proposed architectures, we can see that the message passing paradigm cannot avoid oversmoothing problem because it does not leverage multi-scale information in each layer and will finally lose local information. An alternate solution to address the oversmoothing problem could be to modify the readout function to $\hat{y} = R(\{h_v^{(0)}, h_v^{(1)}, \dots, h_v^{(T)} | v \in \mathcal{V}\})$.

## 5 Experiments

On node classification tasks, we test 2 instances of the snowball GCN and 1 instance of the truncated Krylov GCN, which include linear snowball GCN ($f = g = $ identity, $p = 1$), snowball GCN ($f = $ Tanh, $g = $ identity, $p = 1$) and truncated Krylov ($f = g = $ Tanh, $p = 0$). The test cases include on public splits [37, 25] of Cora, Citeseer and PubMed[2], as well as

Table 1: Algorithms in Matrix and Nodewise Forms

| Algorithms | Forms | |
|---|---|---|
| | Matrix | Nodewise |
| Message Passing | $m^{(t+1)} = M_t(A, h^{(t)})$ <br> $h^{(t+1)} = U_t(h^{(t)}, m^{(t+1)})$ | $m_v^{(t+1)} = \sum_{w \in N(v)} M_t(h_v^{(t)}, h_w^{(t)}, e_{vw})$ <br> $h_v^{(t+1)} = U_t(h_v^{(t)}, m_v^{(t+1)})$ |
| GraphSAGE-GCN | $m^{(t+1)} = Lh^{(t)}$ <br> $h^{(t+1)} = \sigma(m^{(t+1)}W_t)$ | $m_v^{(t+1)} = \text{mean}(\{h_v^{(t)}\} \cup \{h_{N(v)}^{(t)}\})$ <br> $h_v^{(t+1)} = \sigma(W_t^T m_v^{(t+1)})$ |
| Snowball | $m^{(t+1)} = L[h^{(0)}\|\dots\|h^{(t)}]$ <br> $h_v^{(t+1)} = \sigma(m^{(t+1)}W_t)$ | $m_v^{(t+1)} = \|_{i=0}^t \text{mean}(\{h_v^{(i)}\} \cup \{h_{N(v)}^{(i)}\})$ <br> $h_v^{(t+1)} = \sigma(W_t^T m_v^{(t+1)})$ |
| Truncated Krylov | $m^{(t+1)} = h^{(t)}\|\dots\|L^{m_t-1}h^{(t)}$ <br> $h^{(t+1)} = \sigma(m^{(t+1)}W_t)$ | $m_v^{(t+1)} = \|_{i=0}^{m_t-1} \text{mean}(\cup_{k=0}^i \{h_{N^k(v)}^{(t)}\})$ <br> $h_v^{(t+1)} = \sigma(W_t^T m_v^{(t+1)})$ |

the crafted smaller splits that are more difficult [25, 21, 31]. We compare the instances against several methods under 2 experimental settings, with or without validations sets. The compared methods with validation sets include graph convolutional networks for fingerprint (GCN-FP) [8], gated graph neural networks (GGNN) [23], diffusion convolutional neural networks (DCNN) [1], Chebyshev networks (Cheby) [7], graph convolutional networks (GCN) [18], message passing neural networks (MPNN) [12], graph sample and aggregate (GraphSAGE) [14], graph partition neural networks (GPNN) [24], graph attention networks (GAT) [33], LanczosNet (LNet) [25] and AdaLanczosNet (AdaLNet) [25]. The copmared methods without validation sets include label propagation using ParWalks (LP) [35], Co-training [21], Self-training [21], Union [21], Intersection [21], GCN without validation [21], Multi-stage training [31], Multi-stage self-supervised (M3S) training [31], GCN with sparse virtual adversarial training (GCN-SVAT) [30] and GCN with dense virtual adversarial training (GCN-DVAT) [30].

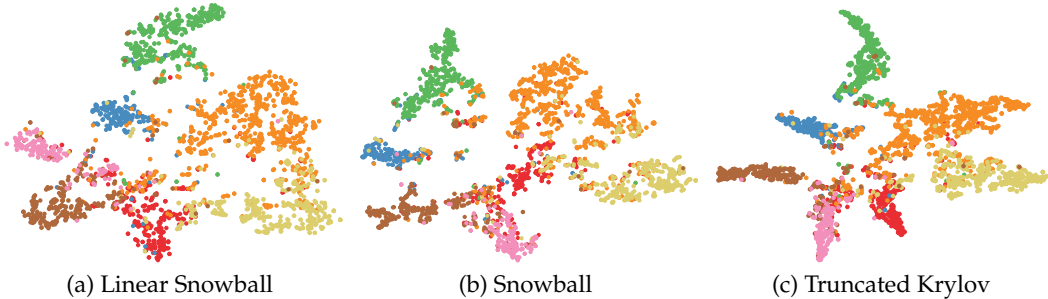

(a) Linear Snowball        (b) Snowball        (c) Truncated Krylov

Figure 3: t-SNE for the extracted features trained on Cora (7 classes) public (5.2%).

In Table 2 and 3, for each test case, we report the accuracy averaged from 10 independent runs using the best searched hyperparameters. These hyperparameters are reported in the appendix, which include learning rate and weight decay for the optimizers RMSprop or Adam for cases with validation or without validation, respectively, taking values in the intervals $[10^{-6}, 5 \times 10^{-3}]$ and $[10^{-5}, 10^{-2}]$, respectively, width of hidden layers taking value in the set $\{100, 200, \cdots, 5000\}$, number of hidden layers in the set $\{1, 2, \dots, 50\}$, dropout in $(0, 0.99]$, and the number of Krylov blocks taking value in $\{1, 2, \dots, 100\}$. An early stopping trick is also used to achieve better training. Specifically we terminate the training after 100 update steps of not improving the training loss.

We see that the instances of the proposed architectures achieve overwhelming performance in *all* test cases. We visualize a representative case using t-SNE [26] in Figure 3. From these visualization, we can see the instances can extract good features with small training

data, especially for the truncated block Krylov network. Particularly, when the training splits are small, they perform astonishingly better than the existing methods. This may be explained by the fact that when there is less labeled data, larger scope of vision field is needed to make recognition of each node or to let the label signals propagate. We would also highlight that the linear snowball GCN can achieve state-of-the-art performance with much less computational cost. If $\mathcal{G}$ has no bipartite components, then in (4), as $n \to \infty$, rank($\boldsymbol{Y'}$) $\leq k$ almost surely.

Table 2: Accuracy without Validation

| Algorithms | Cora | | | | | | CiteSeer | | | | | | PubMed | | | |
|---|---|---|---|---|---|---|---|---|---|---|---|---|---|---|---|---|
| | 0.5% | 1% | 2% | 3% | 4% | 5% | 0.5% | 1% | 2% | 3% | 4% | 5% | 0.03% | 0.05% | 0.1% | 0.3% |
| LP | 56.4 | 62.3 | 65.4 | 67.5 | 69.0 | 70.2 | 34.8 | 40.2 | 43.6 | 45.3 | 46.4 | 47.3 | 61.4 | 66.4 | 65.4 | 66.8 |
| Cheby | 38.0 | 52.0 | 62.4 | 70.8 | 74.1 | 77.6 | 31.7 | 42.8 | 59.9 | 66.2 | 68.3 | 69.3 | 40.4 | 47.3 | 51.2 | 72.8 |
| Co-training | 56.6 | 66.4 | 73.5 | 75.9 | 78.9 | 80.8 | 47.3 | 55.7 | 62.1 | 62.5 | 64.5 | 65.5 | 62.2 | 68.3 | 72.7 | 78.2 |
| Self-training | 53.7 | 66.1 | 73.8 | 77.2 | 79.4 | 80.0 | 43.3 | 58.1 | 68.2 | 69.8 | 70.4 | 71.0 | 51.9 | 58.7 | 66.8 | 77.0 |
| Union | 58.5 | 69.9 | 75.9 | 78.5 | 80.4 | 81.7 | 46.3 | 59.1 | 66.7 | 66.7 | 67.6 | 68.2 | 58.4 | 64.0 | 70.7 | 79.2 |
| Intersection | 49.7 | 65.0 | 72.9 | 77.1 | 79.4 | 80.2 | 42.9 | 59.1 | 68.6 | 70.1 | 70.8 | 71.2 | 52.0 | 59.3 | 69.7 | 77.6 |
| MultiStage | 61.1 | 63.7 | 74.4 | 76.1 | 77.2 | | 53.0 | 57.8 | 63.8 | 68.0 | 69.0 | | 57.4 | 64.3 | 70.2 | |
| M3S | 61.5 | 67.2 | 75.6 | 77.8 | 78.0 | | 56.1 | 62.1 | 66.4 | 70.3 | 70.5 | | 59.2 | 64.4 | 70.6 | |
| GCN | 42.6 | 56.9 | 67.8 | 74.9 | 77.6 | 79.3 | 33.4 | 46.5 | 62.6 | 66.9 | 68.7 | 69.6 | 46.4 | 49.7 | 56.3 | 76.6 |
| GCN-SVAT | 43.6 | 53.9 | 71.4 | 75.6 | 78.3 | 78.5 | 47.0 | 52.4 | 65.8 | 68.6 | 69.5 | 70.7 | 52.1 | 56.9 | 63.5 | 77.2 |
| GCN-DVAT | 49 | 61.8 | 71.9 | 75.9 | 78.4 | 78.6 | 51.5 | 58.5 | 67.4 | 69.2 | 70.8 | 71.3 | 53.3 | 58.6 | 66.3 | 77.3 |
| *linear Snowball* | 67.6 | 74.6 | 78.9 | 80.9 | 82.3 | 82.9 | 56.0 | 63.4 | 69.3 | 70.6 | 72.5 | 72.6 | 65.5 | 68.5 | 73.6 | 79.7 |
| *Snowball* | 68.4 | 73.2 | 78.4 | 80.8 | 82.3 | 83.0 | 56.4 | 63.9 | 68.7 | 70.5 | 71.8 | 72.8 | 66.5 | 68.6 | 73.2 | **80.1** |
| *truncated Krylov* | **71.8** | **76.5** | **80.0** | **82.0** | **83.0** | **84.1** | **59.9** | **66.1** | **69.8** | **71.3** | **72.3** | **73.7** | **68.7** | **71.4** | **75.5** | **80.4** |

For each (column), the greener the cell, the better the performance. The redder, the worse. If our methods achieve better performance than all others, the corresponding cell will be in bold.

Table 3: Accuracy with Validation

| Algorithms | Cora | | | | CiteSeer | | | PubMed | | | |
|---|---|---|---|---|---|---|---|---|---|---|---|
| | 0.5% | 1% | 3% | 5.2% public | 0.5% | 1% | 3.6% public | 0.03% | 0.05% | 0.1% | 0.3% public |
| Cheby | 33.9 | 44.2 | 62.1 | 78.0 | 45.3 | 59.4 | 70.1 | 45.3 | 48.2 | 55.2 | 69.8 |
| GCN-FP | 50.5 | 59.6 | 71.7 | 74.6 | 43.9 | 54.3 | 61.5 | 56.2 | 63.2 | 70.3 | 76.0 |
| GGNN | 48.2 | 60.5 | 73.1 | 77.6 | 44.3 | 56.0 | 64.6 | 55.8 | 63.3 | 70.4 | 75.8 |
| DCNN | 59.0 | 66.4 | 76.7 | 79.7 | 53.1 | 62.2 | 69.4 | 60.9 | 66.7 | 73.1 | 76.8 |
| MPNN | 46.5 | 56.7 | 72.0 | 78.0 | 41.8 | 54.3 | 64.0 | 53.9 | 59.6 | 67.3 | 75.6 |
| GraphSAGE | 37.5 | 49.0 | 64.2 | 74.5 | 33.8 | 51.0 | 67.2 | 45.4 | 53.0 | 65.4 | 76.8 |
| GAT | 41.4 | 48.6 | 56.8 | 83.0 | 38.2 | 46.5 | 72.5 | 50.9 | 50.4 | 59.6 | 79.0 |
| GCN | 50.9 | 62.3 | 76.5 | 80.5 | 43.6 | 55.3 | 68.7 | 57.9 | 64.6 | 73.0 | 77.8 |
| LNet | 58.1 | 66.1 | 77.3 | 79.5 | 53.2 | 61.3 | 66.2 | 60.4 | 68.8 | 73.4 | 78.3 |
| AdaLNet | 60.8 | 67.5 | 77.7 | 80.4 | 53.8 | 63.3 | 68.7 | 61.0 | 66.0 | 72.8 | 78.1 |
| *linear Snowball* | 72.5 | 76.3 | 82.2 | 83.3 | 62.0 | 66.7 | 72.9 | 70.8 | 72.1 | 75.6 | 79.1 |
| *Snowball* | 71.2 | 76.6 | 81.9 | 83.2 | 61.0 | 66.4 | 73.3 | 69.9 | 72.7 | 75.2 | 79.2 |
| *truncated Krylov* | 74.8 | 78.0 | 82.7 | 83.2 | 64.0 | 68.3 | 73.9 | 72.2 | 74.9 | 78.0 | 80.1 |

## 6 Future Works

Future research of this like includes: 1) Investigating how the pointwise nonlinear activation functions influence block vectors, *e.g.*, the feature block vector $\boldsymbol{X}$ and hidden feature block vectors $\boldsymbol{H_i}$, so that we can find possible activation functions better than Tanh; 2) Finding a better way to leverage the block Krylov algorithms instead of conducting simple truncation.

## Acknowledgements

The authors wish to express sincere gratitude for the computational resources of Compute Canada provided by Mila, as well as for the proofreading done by Sitao and Mingde's good friend & coworker Ian P. Porada.

## Footnotes

[1]The expressive power of a sound deep NN architecture should be expected to grow with the increment of network depth [19, 16].

[2]Source code to be found at https://github.com/PwnerHarry/Stronger_GCN

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
