[Supplementary Material]

# Appendices

## A Proofs of Theorems 1 and 2

**Lemma 1.** Suppose that a graph $\mathcal{G}$ has $k$ connected components $\{C_i\}_{i=1}^k$ and $L$ is diffusion operator defined in (3). If $\mathcal{G}$ has no bipartite components, then $\lambda_i(L) \in (-1, 1]$ with

$$\lambda_1 = \cdots = \lambda_k = 1 > |\lambda_{k+1}| \geq \cdots \geq |\lambda_N|$$

*Proof* See Theorem 1 in [21]. □

**Theorem 1.** Suppose that $\mathcal{G}$ has $k$ connected components and the diffusion operator $L$ is defined as that in (3). Let $X \in \mathbb{R}^{N \times F}$ be any block vector and let $W_j$ be any non-negative parameter matrix with $\|W_j\|_2 \leq 1$ for $j = 0, 1, \ldots$. If $\mathcal{G}$ has no bipartite components, then in (4), as $n \to \infty$, $\text{rank}(Y') \leq k$.

*Proof* Note that $Y'$ is $N$ by $F$. Certainly $\text{rank}(Y') \leq k$ if $k \geq F$. In the following we assume $k < F$.

Let $Y_0 = \text{ReLU}(LXW_0)$, then $Y_0$ is a non-negative block vector. Since $L$ and $W_1$ are non-negative as well, we have

$$L\text{ReLU}(LY_0W_1)W_2 = LLY_0W_1W_2 = L^2Y_0W_1W_2$$

which is non-negative. In general, it is easy to see from (4), we have

$$Y' = L^n Y_0 W_1 W_2 \cdots W_n$$

Thus, with the condition $\|W_j\|_2 \leq 1$ for any $j$, the $i$-th largest singular value of $Y'$ satisfies

$$\sigma_i(Y') \leq \sigma_i(L^n)\|Y_0\|_2\|W_1\|_2 \cdots \|W_n\|_2 \leq |\lambda_i(L)|^n \|Y_0\|_2, \quad i = 1, 2, \ldots, \min\{N, F\}$$

From Lemma 1 we can conclude that

$$\lim_{n \to \infty} \sigma_i(Y') = 0, \quad i = k+1, k+2, \ldots, \min\{N, F\}$$

Thus, $\lim_{n \to \infty} \text{rank}(Y') \leq k$. □

**Theorem 2.** Suppose the $n$-dimensional $x$ and $y$ are independently sampled from a continuous distribution and the activation function $\text{Tanh}(z) = \frac{e^z - e^{-z}}{e^z + e^{-z}}$ is applied to $[x, y]$ pointwisely, then

$$\mathbb{P}(\text{rank}(\text{Tanh}([x, y])) = \text{rank}([x, y])) = 1$$

*Proof* Since $x$ and $y$ are sampled from a continuous distribution, $\mathbb{P}(\text{rank}([x, y]) = 2) = 1$ (see [9]). Then

$$\mathbb{P}(\text{rank}(\text{Tanh}([x, y])) = \text{rank}([x, y]))$$
$$= \mathbb{P}(\text{rank}(\text{Tanh}([x, y])) = \text{rank}([x, y]) \,|\, \text{rank}([x, y]) = 2)\mathbb{P}(\text{rank}([x, y]) = 2)$$
$$+ \mathbb{P}(\text{rank}(\text{Tanh}([x, y])) = \text{rank}([x, y]) \,|\, \text{rank}([x, y]) < 2)\mathbb{P}(\text{rank}([x, y]) < 2)$$
$$= \mathbb{P}(\text{rank}(\text{Tanh}([x, y])) = \text{rank}([x, y]) \,|\, \text{rank}([x, y]) = 2) \tag{11}$$

For any fixed $x \in \mathbb{R}^n$, suppose $x$ and random $y$ are linearly independent, but $\text{Tanh}(x)$ and $\text{Tanh}(y)$ are linearly dependent. Without loss of generality, we assume $x_n \neq 0$. Thus $\text{Tanh}(x_n) \neq 0$ and $\text{Tanh}(x_n) \neq 0$. Then we have

$$\frac{\text{Tanh}(y_i)}{\text{Tanh}(y_n)} = \frac{\text{Tanh}(x_i)}{\text{Tanh}(x_n)}, \quad i = 2, \ldots, n$$

Thus,

$$y_i = \text{Tanh}^{-1}\left(\frac{\text{Tanh}(x_i)\text{Tanh}(y_n)}{\text{Tanh}(x_n)}\right), \quad i = 2, \ldots, n$$

For any fixed $x$, the set formed by all $y$ satisfying the above equalities has dimension 1, and therefore its Lebesgue measure is 0, implying that

$$\mathbb{P}(\text{rank}(\text{Tanh}([x, y])) = 1 \,|\, \text{rank}([x, y]) = 2) = 0$$

Then from (11) we can conclude the result holds. □

# B  Numerical Experiments on Synthetic Data

The goal of the experiments is to test which network structure with which kind of activation function has the potential to be extended to deep architecture. We measure this potential by the numerical rank of the output features in each hidden layer of the networks using synthetic data. The reason of choosing this measure can be explained by Theorem 2.2. We build the certain networks with depth 100 and the data is generated as follows.

We first randomly generate edges of an Erdős-Rényi graph $G(1000, 0.01)$, *i.e.* the existence of the edge between any pair of nodes is a Bernoulli random variable with $p = 0.01$. Then, we construct the corresponding adjacency matrix $A$ of the graph which is a $\mathbb{R}^{1000 \times 1000}$ matrix. We generate a $\mathbb{R}^{1000 \times 500}$ feature matrix $X$ and each of its element is drawn from $N(0, 1)$. We normalize $A$ and $X$ as [18] and abuse the notation $A, X$ to denote the normalized matrices. We keep 3 blocks in each layer of truncated block Krylov network. The number of input channel in each layer depends on the network structures and the number of output channel is set to be 128 for all networks. Each element in every parameter matrix $W_i$, $i = 1, \ldots, 100$ is randomly sampled from $N(0, 1)$ and the size is $\mathbb{R}^{\#input \times \#output}$. With the synthetic $A, X, W_i$, we simulate the feedforward process according to the network architecture and collect the numerical rank (at most 128) of the output in each of the 100 hidden layers. For each activation function under each network architecture, we repeat the experiments for 20 times and plot the mean results with standard deviation bars.

# C  Rank Comparison of Activation Functions and Networks

| (a) GCN | (b) Snowball | (c) Truncated Block Krylov |

Figure 4: Column ranks of different activation functions with the same architecture

| (a) ReLU | (b) Identity | (c) Tanh |

Figure 5: Column ranks of different architectures with the same activation function

|          |             |           |
|:--------:|:-----------:|:---------:|
| (a) Cora | (b) Citeseer | (c) PubMed |

Figure 6: Spectrum of the renormalized adjacency matrices for several datasets

## D  Spectrum of the Datasets

## E  Experiment Settings and Hyperparameters

The so-called public splits in [25] and the setting that randomly sample 20 instances for each class as labeled data in [37] is actually the same. Most of the results for the algorithms with validation are cited from [25], where they are reproduced with validation. However, some of them actually do not use validation in original papers and can achieve better results. In the paper, We compare with their best results.

We use NVIDIA apex amp mixed-precision plugin for PyTorch to accelerate our experiments. Most of the results were obtained from NVIDIA V100 clusters on Beluga of Compute-Canada, with minor part of them obtained from NVIDIA K20, K80 clusters on Helios Compute-Canada. The hyperparameters are searched using Bayesian optimization.

A useful tip is the smaller your training set is, the larger dropout probability should be set and the larger early stopping you should have.

Table 5 and Table 4 show the hyperparameters to achieve the performance in the experiments, for cases without and with validation, respectively. When conducting the hyperparameter search, we encounter memory problems: current GPUs cannot afford deeper and wider structures. But we do observe better performance with the increment of the network size. It is expected to achieve better performance with more advanced deep learning devices.

Table 4: Hyperparameters for Tests with Validation

| Architecture | Dataset | Split | Accuracy Our Best | SOTA | learning rate | weight decay | width | depth/blocks | dropout | optimizer |
|---|---|---|---|---|---|---|---|---|---|---|
| **linear Snowball** | **Cora** | 0.5% | 72.51 | 60.8 | 1.8914E-03 | 9.1551E-03 | 4800 | 3 | 0.98369 | RMSprop |
| | | 1% | 76.32 | 67.5 | 2.0050E-03 | 5.0915E-03 | 1200 | 4 | 0.96848 | RMSprop |
| | | 3% | 82.24 | 77.7 | 4.3412E-03 | 2.1344E-03 | 900 | 2 | 0.98323 | RMSprop |
| | | 5.2% (public) | 83.26 | 83.0 | 2.5363E-05 | 1.2692E-02 | 4100 | 3 | 0.63953 | RMSprop |
| | **CiteSeer** | 0.5% | 62.03 | 53.8 | 1.9738E-03 | 1.9239E-02 | 2200 | 2 | 0.98915 | RMSprop |
| | | 1% | 66.71 | 63.3 | 1.0737E-03 | 2.4510E-02 | 800 | 3 | 0.97069 | RMSprop |
| | | 3.6% (public) | 72.85 | 72.5 | 4.5256E-03 | 7.4001E-03 | 4100 | 1 | 0.86582 | RMSprop |
| | **Pubmed** | 0.03% | 70.81 | 61.0 | 2.8443E-04 | 3.4670E-02 | 200 | 10 | 0.98961 | RMSprop |
| | | 0.05% | 72.14 | 68.8 | 3.9460E-03 | 4.6622E-02 | 100 | 4 | 0.8315 | RMSprop |
| | | 0.1% | 75.60 | 73.4 | 2.4167E-03 | 7.4730E-03 | 100 | 5 | 0.93811 | RMSprop |
| | | 0.3% (public) | 79.10 | 79.0 | 3.9812E-03 | 2.1414E-02 | 400 | 3 | 0.96498 | RMSprop |
| **Snowball** | **Cora** | 0.5% | 71.20 | 60.8 | 1.5666E-05 | 1.0674E-02 | 500 | 19 | 0.56764 | RMSprop |
| | | 1% | 76.63 | 67.5 | 2.2739E-04 | 3.4224E-02 | 200 | 14 | 0.76807 | RMSprop |
| | | 3% | 81.88 | 77.7 | 7.6164E-05 | 6.0082E-03 | 200 | 21 | 0.80589 | RMSprop |
| | | 5.2% (public) | 83.19 | 83.0 | 7.7121E-05 | 3.2939E-02 | 4900 | 3 | 0.79489 | RMSprop |
| | **CiteSeer** | 0.5% | 61.03 | 53.8 | 1.0054E-03 | 4.2595E-02 | 1900 | 3 | 0.97837 | RMSprop |
| | | 1% | 66.36 | 63.3 | 3.8615E-04 | 4.1289E-02 | 1300 | 5 | 0.93554 | RMSprop |
| | | 3.6% (public) | 73.32 | 72.5 | 2.5530E-03 | 1.4541E-02 | 3700 | 1 | 0.98481 | RMSprop |
| | **Pubmed** | 0.03% | 69.91 | 61.0 | 6.1538E-03 | 3.5248E-02 | 200 | 8 | 0.45679 | RMSprop |
| | | 0.05% | 72.67 | 68.8 | 4.0294E-03 | 3.2839E-03 | 100 | 18 | 0.81272 | RMSprop |
| | | 0.1% | 75.16 | 73.4 | 2.3525E-03 | 1.5485E-03 | 3200 | 1 | 0.93519 | RMSprop |
| | | 0.3% (public) | 79.16 | 79.0 | 9.4770E-03 | 8.8894E-04 | 1500 | 1 | 0.97378 | RMSprop |
| **truncated Krylov** | **Cora** | 0.5% | 74.78 | 60.8 | 2.5929E-03 | 4.4878E-04 | 100 | 89 | 0.9568 | RMSprop |
| | | 1% | 78.05 | 67.5 | 4.3995E-03 | 9.0436E-05 | 1200 | 40 | 0.96778 | RMSprop |
| | | 3% | 82.67 | 77.7 | 8.2278E-03 | 3.6505E-04 | 2200 | 26 | 0.98803 | RMSprop |
| | | 5.2% (public) | 83.16 | 83.0 | 5.9441E-04 | 6.8103E-03 | 900 | 54 | 0.9018 | RMSprop |
| | **CiteSeer** | 0.5% | 64.04 | 53.8 | 8.0455E-03 | 2.0007E-03 | 100 | 44 | 0.82302 | RMSprop |
| | | 1% | 68.26 | 63.3 | 4.5515E-03 | 7.2589E-03 | 100 | 26 | 0.90988 | RMSprop |
| | | 3.6% (public) | 73.86 | 72.5 | 3.8171E-03 | 1.4549E-02 | 400 | 14 | 0.9857 | RMSprop |
| | **Pubmed** | 0.03% | 72.17 | 61.0 | 6.9072E-03 | 1.1979E-03 | 3300 | 20 | 0.97751 | RMSprop |
| | | 0.05% | 74.89 | 68.8 | 7.8567E-03 | 7.8038E-04 | 2400 | 25 | 0.90062 | RMSprop |
| | | 0.1% | 77.97 | 73.4 | 1.5593E-03 | 6.3401E-03 | 4000 | 22 | 0.97544 | RMSprop |
| | | 0.3% (public) | 80.12 | 79.0 | 8.3614E-04 | 8.4003E-03 | 4300 | 18 | 0.17299 | RMSprop |

Table 5: Hyperparameters for Tests without Validation

| Architecture | Dataset | Split | Accuracy | | Corresponding Hyperparameters | | | | | |
|---|---|---|---|---|---|---|---|---|---|---|
| | | | Our Best | SOTA | learning rate | weight decay | width | depth/blocks | dropout | optimizer |
| **linear Snowball** | Cora | 0.5% | 67.575 | 61.5 | 6.1182E-04 | 4.9810E-03 | 600 | 7 | 0.62185 | Adam |
| | | 1% | 74.579 | 69.9 | 1.6250E-04 | 7.4574E-04 | 200 | 20 | 0.43624 | Adam |
| | | 2% | 78.921 | 75.9 | 8.3569E-04 | 2.0128E-03 | 1700 | 3 | 0.98032 | Adam |
| | | 3% | 80.874 | 78.5 | 1.2211E-04 | 1.6298E-03 | 3600 | 3 | 0.98201 | Adam |
| | | 4% | 82.308 | 80.4 | 9.9572E-05 | 6.3827E-03 | 2100 | 4 | 0.97283 | Adam |
| | | 5% | 82.932 | 81.7 | 7.1942E-06 | 1.7084E-02 | 1400 | 7 | 0.15156 | Adam |
| | CiteSeer | 0.5% | 55.957 | 56.1 | 1.2813E-04 | 2.4846E-02 | 1300 | 5 | 0.12132 | Adam |
| | | 1% | 63.4 | 62.1 | 4.9496E-03 | 5.0868E-03 | 800 | 2 | 0.10184 | Adam |
| | | 2% | 69.251 | 68.6 | 5.0141E-04 | 2.8694E-02 | 2400 | 3 | 0.95313 | Adam |
| | | 3% | 70.635 | 70.3 | 5.1388E-04 | 3.5018E-02 | 2100 | 3 | 0.96181 | Adam |
| | | 4% | 72.48 | 70.8 | 7.3531E-05 | 4.5418E-02 | 2700 | 3 | 0.37424 | Adam |
| | | 5% | 72.639 | 71.3 | 2.1794E-04 | 4.9282E-02 | 3200 | 3 | 0.86498 | Adam |
| | Pubmed | 0.03% | 65.479 | 62.2 | 8.8680E-04 | 3.3575E-02 | 400 | 9 | 0.21978 | Adam |
| | | 0.05% | 68.523 | 68.3 | 1.1179E-03 | 2.5143E-02 | 400 | 7 | 0.34326 | Adam |
| | | 0.1% | 73.588 | 72.7 | 4.6872E-04 | 7.8163E-03 | 900 | 5 | 0.19117 | Adam |
| | | 0.3% | 79.691 | 79.2 | 2.2653E-04 | 2.9657E-03 | 3400 | 4 | 0.98996 | Adam |
| **Snowball** | Cora | 0.5% | 68.425 | 61.5 | 6.7929E-04 | 4.5636E-03 | 100 | 22 | 0.00547 | Adam |
| | | 1% | 73.152 | 69.9 | 1.6805E-03 | 9.9231E-04 | 3200 | 2 | 0.90518 | Adam |
| | | 2% | 78.405 | 75.9 | 1.3363E-05 | 4.5665E-04 | 500 | 16 | 0.3652 | Adam |
| | | 3% | 80.827 | 78.5 | 1.9982E-04 | 2.4818E-02 | 4300 | 3 | 0.41048 | Adam |
| | | 4% | 82.303 | 80.4 | 1.9945E-04 | 6.0539E-03 | 2300 | 2 | 0.96809 | Adam |
| | | 5% | 83.006 | 81.7 | 3.2402E-04 | 1.3194E-02 | 3800 | 3 | 0.17131 | Adam |
| | CiteSeer | 0.5% | 56.438 | 56.1 | 4.6535E-05 | 2.1550E-02 | 1400 | 6 | 0.00548 | Adam |
| | | 1% | 63.862 | 62.1 | 1.4755E-03 | 2.8593E-02 | 2100 | 4 | 0.92137 | Adam |
| | | 2% | 68.729 | 68.6 | 3.1813E-05 | 2.2883E-02 | 2200 | 2 | 0.15915 | Adam |
| | | 3% | 70.534 | 70.3 | 3.2765E-05 | 2.4819E-02 | 2300 | 4 | 0.88698 | Adam |
| | | 4% | 71.813 | 70.8 | 3.8585E-04 | 3.6265E-02 | 2300 | 2 | 0.30763 | Adam |
| | | 5% | 72.806 | 71.3 | 7.1685E-05 | 4.9615E-02 | 3900 | 3 | 0.87297 | Adam |
| | Pubmed | 0.03% | 66.477 | 62.2 | 4.9118E-05 | 1.6182E-03 | 200 | 22 | 0.028551 | Adam |
| | | 0.05% | 68.583 | 68.3 | 1.1521E-03 | 3.8871E-02 | 400 | 7 | 0.059136 | Adam |
| | | 0.1% | 73.194 | 72.7 | 5.1533E-04 | 1.2711E-02 | 3500 | 2 | 0.98885 | Adam |
| | | 0.3% | 80.14 | 79.2 | 4.3710E-05 | 3.9694E-02 | 5000 | 2 | 0.067568 | Adam |
| **truncated Krylov** | Cora | 0.5% | 71.819 | 61.5 | 1.0652E-04 | 3.0371E-04 | 100 | **97** | 0.47949 | Adam |
| | | 1% | 76.485 | 69.9 | 4.3309E-03 | 2.3969E-04 | 300 | 25 | 0.96104 | Adam |
| | | 2% | 79.974 | 75.9 | 9.9421E-04 | 6.4090E-04 | 1400 | 33 | 0.8084 | Adam |
| | | 3% | 82.047 | 78.5 | 4.9624E-03 | 1.3848E-04 | 700 | 16 | 0.98362 | Adam |
| | | 4% | 82.965 | 80.4 | 2.1988E-03 | 3.9724E-04 | 100 | 70 | 0.81721 | Adam |
| | | 5% | 84.109 | 81.7 | 6.8068E-03 | 3.2025E-04 | 500 | 18 | 0.97897 | Adam |
| | CiteSeer | 0.5% | 59.85 | 56.1 | 4.8252E-03 | 2.1583E-03 | 500 | 22 | 0.98663 | Adam |
| | | 1% | 66.073 | 62.1 | 1.7210E-03 | 1.9423E-03 | 100 | 27 | 0.74055 | Adam |
| | | 2% | 69.809 | 68.6 | 6.4732E-03 | 4.2307E-03 | 400 | 11 | 0.16691 | Adam |
| | | 3% | 71.3 | 70.3 | 5.8873E-04 | 2.0091E-02 | 1400 | 11 | 0.39397 | Adam |
| | | 4% | 72.343 | 70.8 | 8.4962E-05 | 4.8571E-02 | 2300 | 7 | 0.70649 | Adam |
| | | 5% | 73.713 | 71.3 | 2.7076E-03 | 1.7906E-02 | 1900 | 9 | 0.70568 | Adam |
| | Pubmed | 0.03% | 68.673 | 62.2 | 7.2129E-05 | 3.3215E-03 | 2500 | 25 | 0.017744 | Adam |
| | | 0.05% | 71.447 | 68.3 | 1.1325E-04 | 2.2466E-03 | 3000 | 23 | 0.98752 | Adam |
| | | 0.1% | 75.539 | 72.7 | 1.9708E-03 | 4.8034E-03 | 3900 | 17 | 0.98818 | Adam |
| | | 0.3% | 80.384 | 79.2 | 1.9555E-03 | 1.4919E-03 | 5000 | 12 | 0.98867 | Adam |