[Reviews · NeurIPS 2019]

Reviewer 1



As described, the paper makes some strong contributions to the knowledge of GCNs. However, some parts of the paper are not clearly written or organized. As an example, the discussion and numerical results around Figure 1 are difficult to understand. Additionally, some figure and table captions and some mathematical derivations are not clearly expressed, and should be revised. The proposed architectures are evaluated and compared to 3 common datasets (all for classification of scientific publications into one of several classes), and the results on these problems are fairly convincing. However, it would be more convincing to expand experiments and interpretation to different families of problems (with diverse data structures), as well as to compare the time and memory complexity associated with these approaches. Apart from the number layers, it is not clear from the paper and its experiments what factors are critical for the performance of the proposed architectures. What key factors can affect the accuracy and complexity of the proposed architectures?

Reviewer 2



# originality The theoretical analyses on the scalability of GCN have great originality and important in practice. The idea to use the Krylov subspace methods to learn the spectral filter has been a sort of universal one recently, e.g., LanczosNet. The proposed architectures, namely, snowball GCN and truncated Krylov block network, have a certain novelty. # quality - I would like to see the ablation test in order to investigate either Tanh or the snowball/truncated Krylov works. This is very important to clarify the source of difficulty to train GCN, given good end-to-end performances. - Experiments mostly follow the procedures in the existing work, which would be fair comparisons. If there are more fancy applications or datasets, that would be more interesting. I guess Tables 1 and 2 do not show precision, rather classification accuracy (I also confirmed the submitted code and it also says the classification accuracy). At least LanczosNet and Multi-stage training report the same numbers in those tables as classification accuracy in their original papers. Ditto to Tables 3 and 4 in the appendix. By chance, I found that the number in LNet + Cora + 3% must be 76.3 if you directly refer to the number in the original paper. - It's interesting to look at two different architectures, while I would like to know why you need to propose two architectures, especially given the equivalence analysis in Section 5.3. Can I say that let's use snowball if you do not have much computational resource, otherwise truncated Krylov? # clarity - The introduction to GCN and the Krylov subspace methods are very good and nice to outsiders. I just suggest a few points that could be improved. * In Section 4.1, the block vector subspace appears in l.92 for the first place, which would be defined later in l.104. Could you make the definition first? * About 5 lines after l.115 (between l.115 and l.116), the spectrum radius is undefined. Could you define it? - The explanation of the motivation for the proposed architectures might be a bit misleading. I might have misunderstood, but let me confirm so as not to get something wrong. You claim that the proposed architectures are motivated by the theoretical analysis (Theorems 1 and 2) in l.39; this is partially the case since you choose to use the Tanh activation instead of Relu, which would make GCN more scalable and retain richer information even in deeper networks. However, the main ideas in your Snowball GCN and Truncated Block Krylov Network seem motivated by how to alleviate the difficulty in the Krylov subspace methods (in Section 4.4). Could you clarify this part? Or you may emphasize this difficulty somehow in the introduction. - Please explain the percentage under dataset names in Tables 1 and 2. (train-validation split ratio, right?) Ditto to Tables 3 and 4 in the appendix. # significance This work gives an important insight that the commonly used activation Relu might not be suitable in case of GCN. The fact is also confirmed through simple empirical studies. The following work can investigate this line as one of the open problems in GCN field. =========== After feedback =========== Thank you for providing answers. I will leave my score the same. I still feel that it is nice to have messages that says which architectures we should use base on some trade-off relationships if any.

Reviewer 3



The paper is clear but some parts could be improved, for example the authors refer to scalability issues for GCN in the sense of stacking multiple layers but the term refers to scalability wrt size of the input. The authors focus on a very specific instantiation of graph convolutional networks, namely GCN, and to spectral methods. How does the method compare to approaches based on the more general message passing paradigm that can implement both local and global computation? Laplacian smoothing is not necessarily an issue there. Why is the graph defined using edges and adjacency? Isn’t it enough to have either one? Please explain. At the end of sec 2 it is said that Chebyshev polynomial constitutes a spectrum-free. The method does not require the computation of the eigendecomposition, however the resulting method still behaves as spectrum-based. Experiments are very thorough and show that the proposed method achieves good performance in all the proposed tasks. However the section is quite short and should be expanded for better clarity. For example, it is not specified what column is the usual data-regime (not decimated setting) used for each experiment. Future works in this current form is not very useful. I’d consider some rewrite of section 4 and 5 to make space for better motivation and explanation of results.

[Author Response · NeurIPS 2019]

## Mutual Concerns of All Reviewers

*"The results around Fig. 1 are difficult to understand; Some figure and table captions and some mathematical derivations should be revised; The experiment section needs to be expanded; Future works in this current form is not very useful."*: We will rewrite the corresponding sections and fix the issues you have pointed out. Thank you all very much!

## Mutual Concerns of Reviewer 2 and Reviewer 3

*"What factors are critical for the performance of the proposed architectures? Ablation tests to see either the network architecture or the activation works is needed."*: The changes in network architecture and in the activation function both contribute to the "scalability" of the network: the ability of increasing the expressive power by enlarging the network. Empirically, such "scalability" is observed for that larger instances of the 2 architectures yield better performance. We have done ablation tests with extensive tuning in each architecture to see their limits. For those best configurations, we observed that the change in activation or architecture alone both contributes to the performance: the architecture contribution is more significant with fewer training labels while that of activation function is the opposite. However, when combined, they did not result in '1+1=2': the contribution of the activation seems being absorbed. Such observation is expected since the number of layers does not suffice to demonstrate significant performance difference also for the characteristics of the tasks. Larger difference with more layers is expected on more complex tasks.

## Mutual Concerns of Reviewer 3 and Reviewer 4

*"Clarify the motivation of the proposed architectures and the necessity of the two different architectures."*:

The oversmoothing problem from which classical GCN suffers, when adding more layers, can be intuitively interpreted that low-level information is neglected in the higher level of information diffusion, as there are no direct connections in the architecture. The two proposed architectures address this issue by stacking levels of information together in **different** manners: Snowball accumulatively stacks those features layer by layer, whereas Truncated Krylov considers all levels of diffusion information simultaneously in each layer.

## Concerns of Reviewer 2 Only

*"More experiments on different tasks, provide a complexity (time and memory) analysis."*: Results on larger datasets and inductive learning tasks will be added, with more results to illustrate the arguments about the activation. The complexity (time and memory) of the new architectures is clearly larger than GCN's due to the dense connected nature of Snowball architecture and the size of the Truncated Krylov networks. As the complexity depends on the sparsity of the matrices (task-dependent) and the mechanisms of pytorch, it is hard to analyze it theoretically. However, we will provide details about the runtime and memory consumptions in the experimental section.

## Concerns of Reviewer 4 Only

*"The authors refer to scalability issues for GCN in the sense of stacking multiple layers but the term refers to scalability wrt size of the input."*: We will state more clearly, *e.g.* the scalability of the size of the network.

*"Why is the graph defined using edges and adjacency? Isn't it enough to have either one?"*: We will fix this.

*"Chebyshev polynomial constitutes a spectrum-free. The method does not require the computation of the eigendecomposition, however the resulting method still behaves as spectrum-based."*: Our original statements aligned with the naive dichotomy of some existing work, where spectrum-free refers also to those behave as spectrum-based with no explicit eigendecomposition. But we do think that your dichotomy is more reasonable. We will make the change.

*"How does the method compare to approaches based on the more general message passing paradigm that can implement both local and global computation? Laplacian smoothing is not necessarily an issue there."*: Denote $N^k(v)$ as the $k-$hop neighbors of node $v$ and $\|$ as concatenation. Message passing paradigm cannot avoid oversmoothing because it does not leverage multi-scale information in each layer. In fact, we need a densely connected architecture. The relations are illustrated in the following table. We can also change the readout function $\hat{y} = R(\{h_v^T, |v \in G\})$ to $\hat{y} = R(\{h_v^0, h_v^1, \ldots, h_v^T, |v \in G\})$ to mitigate oversmoothing.

|  | Massage Passing | GraphSAGE-GCN | Snowball | Truncated Krylov |
|---|---|---|---|---|
| Matrix | $m^{t+1} = M_t(A, h^t)$ <br> $h^{t+1} = U_t(h^t, m^{t+1})$ | $m^{t+1} = Lh^t$ <br> $h^{t+1} = \sigma(m^{t+1}W^t)$ | $m^{t+1} = L[h^0\|\ldots\|h^t]$ <br> $h_v^{t+1} = \sigma(m^{t+1}W^t)$ | $m^{t+1} = h^t\|\ldots\|L^{m_t-1}h^t$ <br> $h^{t+1} = \sigma(m^{t+1}W^t)$ |
| Nodewise | $m_v^{t+1} = \sum_{w \in N(v)} M_t(h_v^t, h_w^t, e_{vw})$ <br> $h_v^{t+1} = U_t(h_v^t, m_v^{t+1})$ | $m_v^{t+1} = \text{mean}(\{h_v^t\} \cup \{h_{N(v)}^t\})$ <br> $h_v^{t+1} = \sigma(W^t m_v^{t+1})$ | $m_v^{t+1} = \|_{i=0}^t \text{mean}(\{h_{N(v)}^i\})$ <br> $h_v^{t+1} = \sigma(W^t m_v^{t+1})$ | $m_v^{t+1} = \|_{i=0}^{m_t-1} \text{mean}(\cup_{k=0}^i \{h_{N^k(v)}^t\})$ <br> $h_v^{t+1} = \sigma(W^t m_v^{t+1})$ |

[Meta-Review · NeurIPS 2019]

A method for performing graph convolutions based on block Krylov subspace forms has been proposed. Two novel GCN architectures (a denseNet-like architecture and stacks truncated Krylov blocks) that make use of the multi-scale information in different ways have been proposed and analysed. The proposed architectures have been evaluated and compared to 3 common datasets (all for classification of scientific publications into one of several classes), and the results are fairly convincing. There is a clear consensus among the reviewers for acceptance. Hence, we recommend acceptance.